# Development and In Vitro–In Vivo Correlation Evaluation of IMM-H014 Extended-Release Tablets for the Treatment of Fatty Liver Disease

**DOI:** 10.3390/ijms241512328

**Published:** 2023-08-02

**Authors:** Chi Zhang, Huihui Shao, Zunsheng Han, Bo Liu, Jing Feng, Jie Zhang, Wenxuan Zhang, Kun Zhang, Qingyun Yang, Song Wu

**Affiliations:** State Key Laboratory of Bioactive Substance and Function of Natural Medicines, Institute of Materia Medica, Chinese Academy of Medical Sciences & Peking Union Medical College, Beijing 100050, China; zczhang@imm.ac.cn (C.Z.); shaohuihui@imm.ac.cn (H.S.); hzs@imm.ac.cn (Z.H.); liuzzbo@163.com (B.L.); fengjing@imm.ac.cn (J.F.); zhd@imm.ac.cn (J.Z.); wxzhang@imm.ac.cn (W.Z.); aimokun@163.com (K.Z.)

**Keywords:** IMM-H014, extended-release tablet, hydrophilic polymers, in vitro–in vivo correlation, non-alcoholic fatty liver disease

## Abstract

This study aimed to develop extended-release tablets containing 25 mg IMM-H014, an original drug formulated by a direct powder pressing method based on pharmaceutical-grade hydrophilic matrix polymers such as hydroxypropyl methylcellulose, to establish an in vitro–in vivo correlation (IVIVC) to predict bioavailability. The tablets’ mechanical properties and in vitro and in vivo performance were studied. The formulation was optimized using a single-factor experiment and the reproducibility was confirmed. The in vitro dissolution profiles of the tablet were determined in five dissolution media, in which the drug released from the hydrophilic tablets followed the Ritger–Peppas model kinetics in 0.01 N HCl medium for the first 2 h, and in phosphate-buffered saline medium (pH 7.5) for a further 24 h. Accelerated stability studies (40 °C, 75% relative humidity) proved that the optimal formulation was stable for 6 months. The in vivo pharmacokinetics study in beagle dogs showed that compared to the IMM-H014 immediate release preparation, the maximum plasma concentration of the extended-release (ER) preparation was significantly decreased, while the maximum time to peak and mean residence time were significantly prolonged. The relative bioavailability was 97.9% based on the area under curve, indicating that the optimal formulation has an obvious ER profile, and a good IVIVC was established, which could be used to predict in vivo pharmacokinetics based on the formulation composition.

## 1. Introduction

As a major global chronic disease, non-alcoholic fatty liver disease (NAFLD) has a disease spectrum that includes simple non-alcoholic fatty liver, non-alcoholic fatty hepatitis, and related liver cirrhosis and liver cell carcinoma [1]. NAFLD is associated with a bleak prognosis, including cirrhosis, hepatocellular carcinoma, and even death [1]. Recent trends in the treatment of NAFLD have focused on lifestyle modification [2], bariatric surgery [3], and medical therapy [4,5]. Given the considerable risk associated with surgery, the exploration of new therapeutic agents against NAFLD is essential but challenging. IMM−H014 (previously known as WS117) is a novel collateral phenyl-structured compound with a new mechanism of action involving a nuclear factor NF−E2-related factor agonist. IMM–H014 has anti-inflammatory activity and can increase insulin sensitivity, which may be useful in the treatment of NAFLD [6].

In the treatment of chronic diseases, IMM-H014 requires long-term administration, and patients with NAFLD require continuous and effective medical care. IMM-H014 is easily absorbed in the gastrointestinal tract, demonstrating a 96.3% oral bioavailability in rats, a short plasma drug concentration peak time (T_max_) (0.5 h), and a short plasma elimination half-life (t_1/2_) (rats, 1.8 h; beagle dogs, 3.5 h) [7]. Drugs with a short half-life are given at shorter intervals and more frequently. To improve patient compliance, it is recommended that IMM-H014 should possess an extended-release (ER) profile after oral administration. ER solid dosage forms for oral administration may be effective for avoiding frequent dosing and may improve long-term therapeutic management with drug molecules that exhibit a narrow therapeutic range and/or are rapidly cleared from the blood [8].

Inert polymer matrices have been widely used as skeleton materials to adjust the release rate in controlled-release delivery formulates [9,10]. The mechanism of action of hydrophilic polymer matrix systems, which are widely used in controlled drug delivery, is based on the gel layer that is formed by hydrating the polymer. The gel layer controls the drug release rate [11,12,13]. The release of water-soluble drugs in vitro controls the diffusion of the gel layer outside, which depends on the gel’s viscosity. In contrast, the release of poorly water-soluble drugs is dependent on the dissolution of the polymer [14,15].

Cellulose derivatives have been widely used as hydrogel matrices for controlled drug delivery, among which, hydroxypropyl methylcellulose (HPMC) is the most widely applied given that it is easy to use, widely available to purchase, and has low/no toxicity [16]. Drug release is controlled by a gel layer formed on the matrix surface due to the hydration of HPMC, through which the loaded drug diffuses [17].

In matrix tablets prepared with HPMC, a gel layer is produced on the surface of the tablets upon aquation. At a higher use level of HPMC, the greater degree of entanglement of the linear polymer chains results in “virtual crosslinking,” leading to the formation of a more robust gel layer [18]. HPMC is a versatile polymer for the production of tablets and is widely accepted as a pharmaceutical excipient for oral administration [19,20]. HPMC can be used to control the release behavior of hydrophilic and hydrophobic drugs through swelling, diffusion, and erosion processes [21].

In this study, we explored the feasibility of producing IMM-H014 extended-release tablets using a direct compression method based on hydrophilic polymeric matrices of HPMC. The effect of the polymer concentration on the in vitro and in vivo drug release rate was researched to establish the preferred formulation in terms of modified release. Furthermore, a point-by-point in vitro–in vivo correlation (IVIVC) was developed to relate the percentage of drug dissolved to the percentage of drug absorbed. The changes in drug absorption in the body can be evaluated by IVIVC based on the in vitro dissolution when the formulation is changed slightly [22].

## 2. Results and Discussion

### 2.1. Solubility Studies

IMM-H014 possesses pH-dependent saturated solubility (as shown in Figure 1 and in Appendix A), as well as high solubility under acidic conditions, as a decreasing solubility was observed with an increasing pH of the medium. Furthermore, IMM-H014 exhibited pH-dependent solubility in different media.

### 2.2. Optimization of IMM−H014 ER Tablet Formulations

#### 2.2.1. Screening of Matrix Materials

First, the type and viscosity of the hydrogel matrix were investigated to achieve the desired ER of IMM-H014. The HPMC and hydroxy propyl cellulose (HPC) used for different tablets are displayed in Table 1 as F1–F4.

As a sulfonate salt, IMM-H014 is unsuitable for wet granulation with ethanol because it could produce methanesulfonate, which has genetic toxicity. However, the hydrogel skeleton material is also unsuitable for wet granulation with water; instead, a suitable process is dry granulation or direct powder pressing. Direct powder pressing is a simple and repeatable process, and can be used as a first choice. HPMC, which can be used for direct pressing, showed better liquidity than HPC, as confirmed by the values of the angle of repose and weight variation (as shown in Table 2). The repose angles when HPMC (90SH–4000SR and 90SH–10,000SR) was used as the skeleton material are 26° and 27°, with a Carr’s Index (CI) of 13.4 and 12.7, respectively. The repose angles when HPC (M-FP and H-FP) was used as the skeleton material are 35° and 37°, with a CI of 17.5 and 18.4. It is generally believed that when the repose angle is ≤30° or the CI is ≤15, the particle flowability is good and can meet the pressing requirements. The results also showed that with the same tablet hardness (100 N–130 N), the weight variation in the tablet was greater when HPC was used as the excipient compared to HPMC.

The tablet made of HPMC (90SH–4000SR) showed ER in vitro which could be completely released after 24 h, with a cumulative release of >95%. The tablets produced with HPMC (90SH–10,000SR) and HPC (M-FP and H-FP) as the skeleton materials were incompletely released (as shown in Appendix A). The research showed that the gel layer formed by the hydrogel matrix was the primary control behind the drug release. The hydration rate of the matrix was related to its viscosity. The gel layer of HPMC (90SH–4000SR) was formed slowly and had a weak strength, resulting in the rapid release of IMM-H014. However, the gel layers formed by HPMC (90SH–10,000SR), HPC ((M-FP), and HPC (H-PC) were formed rapidly and had a high strength, preventing the release of IMM−H014. Therefore, HPMC (90SH–4000SR) was used as the skeleton material of the IMM-H014 tablet.

#### 2.2.2. HPMC Concentration

Next, the concentration of HPMC was analyzed (as shown in Table 1; IR, F5–F10) to achieve the ideal release effect of the IMM-H014 ER tablets. The amount of HPMC had a significant effect on the release profile, while the physical properties of F5–F10 demonstrated that the concentration of hydrophilic polymers did not influence the physical properties of the obtained granules and tablets, such as the angle of repose, CI, hardness of tablets, and weight variation in tablets (as shown in Table 2).

During the in vitro release study, the IMM–H014 tablets remained intact, although a gel layer formed around them. The in vitro dissolution results showed that the in vitro release rate increased with the decrease in the ratio of HPMC in the formulation. The cumulative release amount of F5 and F10 was 19.1% and 71.9% in 1 h, respectively, while the cumulative immediate release (IR) was 100% in <1 h (as shown in Figure 2 and Appendix A).

The effect of different HPMC dosages on the pharmacokinetics in beagle dogs showed that with the increase in the amount of the matrix material, the release rate of the IMM-H014 tablets in vitro, the absorption in vivo, and the absorption amount all decreased. When the amount of the skeleton material was high, the bioavailability in beagles was low and could not be fully absorbed. When the dosage of the skeleton material (HPMC) was 15% (F9), the area under the curve (AUC) of the ER IMM–H014 tablet was equivalent to that of the IR preparation (as shown in Table 1), the mean residence time (MRT) was significantly lengthened, the maximum plasma concentration (C_max_) was significantly lowered, and the in vivo release was slower (as shown in Figure 3 and Appendix A). It has been previously shown that diffusion is a key factor affecting the release of IMM-H014. With the increase in HPMC concentration, the thickness of the gel layer formed around the sustained-release tablets increases, as does the diffusion path. The mechanical properties of the polymer network were improved by creating a longer diffusion path, thus reducing the drug release from the tablets.

### 2.3. In Vitro Dissolution Studies

Next, to study the dissolution behavior of the best formulation (F9), an in vitro dissolution study was performed for the IMM-H014 ER tablets in the following five dissolution media: 0.01 N HCl, pH 4.5 acetate buffer extraction procedure (ABEP), pH 6.8 phosphate-buffered saline (PBS), water, and pH 2.0 → pH 7.5 PBS (as shown in Figure 4); the total amount released within 24 h reached more than 90% in all five media. The optimal formulation (F9) showed good ER characteristics in the five dissolution media. Notably, the in vitro release behavior of the optimal IMM–H014 ER tablets was affected by the pH of the dissolution media to some extent. The drug release amount within the initial stage (1 h) under the condition of a pH of 2.0 was higher than those in other media. As the pH of the medium increased, the release rate decreased, which was consistent with the finding that IMM–H014 possesses pH-dependent saturated solubility.

### 2.4. Drug Release Mechanism Studies

To elucidate the release kinetic characteristics of IMM–H014 ER tablets in pH 2.0 to pH 7.5 media, The zero−order, first-order, Higuchi, and Ritger–Peppas models were used to accommodate the drug release kinetics model (as shown in Table 3). The fitting result was Q_t_ = 32.87 t^0.45^, and the corresponding R^2^ was 0.97521, which showed a good and reasonable correlation with the Ritger–Peppas model. Moreover, the *n* value was 0.45, which confirmed that IMM-H014 was released from the optimized formulation by diffusion, which was consistent with the high solubility of IMM-H014 in an aqueous solution.

### 2.5. Reproducibility

For their further application in industry, the reproducibility of the production processes of the optimal IMM-H014 ER tablets was explored. Three batches each comprising 20,000 tablets were prepared. The results showed no pronounced differences in the visual appearance of three batches of products, and the IMM–H014 content was >99% in all three batches. Moreover, the in vitro release profile (as shown in Table 4, and in Appendix A) showed that the calculated f_2_ values among the different batches were all greater than 50 (Table 4). Reproducibility and rationality were elucidated by the acceptable S.D. and similar drug release profiles.

### 2.6. Stability Evaluation

The appearance, content, and in vitro release profile of the optimal IMM-H014 ER tablets were studied after the tablets were reposited for 6 months under accelerated circumstances (40 °C, relative humidity (RH) 75% ± 5%) (as shown in Table 5). The tablets showed no obvious changes in terms of their appearance and content over the 6-month period. Additionally, the f_2_ values of the IMM–H014 ER tablets in the five media were still higher than 50 after 6 months of storage, suggesting a similar release behavior compared to that of the 0-month formulation.

### 2.7. In Vivo Pharmacokinetic Studies

The in vivo pharmacokinetic outlines and calculated pharmacokinetic parameters after oral administration of IMM-H014 ER and IR to beagles are shown in Figure 5 and Table 6. The IMM-H014 content in plasma was analyzed at different points. As expected, compared to IR, the plasma concentration of the IMM–H014 ER tablets was more stable. The T_max_ of IR and ER tablets was 0.7 h and 2.5 h, respectively, demonstrating the delayed T_max_ of IMM-H014. The C_max_ of the plasmatic IMM-H014 concentration for the IR and ER tablets was 3106.894 ng/mL and 1067.956 ng/mL, respectively, demonstrating the obviously decreased C_max_ of IMM-H014. The values of MRT for IR and ER were 6.1 h and 9.8 h, respectively, indicating the slower elimination of the IMM-H014 ER tablets. Meanwhile, there was a significant difference in the T_max_, C_max_, and MRT between the IMM–H014 IR and ER tablets (*p* < 0.05). Additionally, the relative bioavailability F of the IMM-H014 ER tablets compared to the IR tablets based on AUC_0→t_ was 97.9%, suggesting that they exhibited similar absorption in the circulation. Compared to IR, the ER tablets showed an obvious ER effect.

### 2.8. In Vivo–In Vitro Correlation

Next, the IVIVC for IMM–H014 was examined. After validation using the DAS2.0 (drug and statistics for Windows) program, the optimization single-compartment model was determined for the drug dosage forms. The percentage of drug release in 0.01 N HCl → pH 7.5 PBS medium of ER tablets and the drug absorption fraction in beagle dogs under a fasted condition were used to investigate the IVIVC. According to the Loo–Riegelman method, the accumulative absorption fraction of IMM–H014 was calculated. As shown in Figure 6, the correlation coefficient (R^2^) between the drug’s in vitro release and in vivo absorption percentage was 0.9509, demonstrating a good association between the drug’s release and fraction absorption. Therefore, in vitro drug release can be used to predicate the in vivo absorption behavior.

## 3. Materials and Methods

### 3.1. Materials

IMM-H014 was produced in the author’s laboratory; IMM-H014 extended-release tablets (batch Nos. 1, 2, and 3) were produced by Pharmaron Ningbo Co., Ltd. (Ningbo, China); hydroxypropyl methylcellulose (HPMC) was purchased from Shin–Etsu Chemical Co., Ltd. (Dalian, China); lactose was purchased from DFE pharma GmbH & Co. KG. (Shanghai, China); silicon dioxide and magnesium stearate were purchased from Anhui Sunhere Pharmaceutical Excipients Co., Ltd. (Huainan, China); and acetonitrile (HPLC grade) and methanol (HPLC grade) were purchased from Honeywell. The other reagents were analytical grade.

Animal: Beagles (female or male, purchased from Beijing Marshall Biotechnology Co., Ltd. Beijing, China) were used for in vivo pharmacokinetic studies. During the experiment, the beagles were given a standard diet and allowed to drink freely. The study followed ethical guidelines and the protocol was approved by the Laboratory Animal Ethics Committee of the Institute of Materia Medica, Chinese Academy of Medical Sciences and Peking Union Medical College (animal ethical clearance protocol number: 00005367, 2022. 12.)

### 3.2. Solubility Test

#### 3.2.1. Preparation Methods for Solubility Media

The saturation solubility of IMM-H014 in pH 2.0, pH 4.5, pH 6.8, and pH 12 media were investigated. These media were prepared as follows:(1)pH 2.0 hydrochloric acid solution: 0.9 mL of hydrochloric acid was diluted with water to 1000 mL.(2)pH 4.5 acetate-buffered solution: 2.99 g of sodium acetate was weighed, to which 1.6 mL glacial acetic acid was added and diluted with water to 1000 mL.(3)pH 6.8 phosphate-buffered solution: 6.8045 g of monopotassium phosphate and 0.896 g of sodium hydroxide were weighed and dissolved in an appropriate amount of water, and the solution was then diluted with water to 1000 mL.(4)pH 12 sodium hydroxide solution: 0.4 g of sodium hydroxide was dissolved with 1000 mL of water.

#### 3.2.2. Determination Method for Solubility Studies

An appropriate amount of the API was added to different pH media at 37.0 ± 0.5 °C, before shaking the flask for 24 h, filtering, and taking the continuous filtrate. The continuous filtrates were then diluted stepwise with the respective media to a suitable concentration within the range of 20–60 μg/mL, and were used as the test solutions. The amount of the dissolved drug was quantified by HPLC, as described in Section 3.6.4.

### 3.3. Production of the Immediate-Release (IR) Capsule of IMM-H014

The appropriate amount of the IMM-H014 active pharmaceutical ingredient (API) was loaded into the gelatin capsule shell as the IMM-H014 immediate-release preparation.

### 3.4. Production of IMM-H014 ER Tablets

IMM-H014 ER tablets were produced by pressing the powder directly. The required quantities of drug and ingredients were passed through an 80-mesh screen (stainless steel). IMM-H014, HPMC, and lactose were mixed completely using the equivalent addition method. Then, after adding silicon dioxide and magnesium, the sample was mixed for a further 10 min. Finally, the tablets were compressed using a flat-faced 8 mm punch. Each tablet contained 35 mg of IMM-H014, with the tablet weight and hardness being maintained at 200 mg and between 100 and 130 N, respectively. The composition of the tablets is shown in Table 6.

### 3.5. Evaluation of Granules

#### 3.5.1. Angle of Repose

The granules’ angle of repose was measured using the funnel method. The granules were precisely weighed and located in the funnel, and the height of the funnel was adjusted so that the cusp of the funnel just touched the top of the granule cone. The granules ran freely through the funnel to the surface. The diameter of the particle cone was evaluated and the angle of repose was determined using Equation (1).
(1)$$Tan\theta =\frac{h}{r}$$
where h is the height of the granule cone and r is the radius of the granule cone.

#### 3.5.2. Bulk Density

Approximately 5.0 g of particles, which was weighed without any aggregation, was placed in a 10.0 mL measuring cylinder to acquire the volume. The cylinder was dropped from a height of 2.5 cm on a flat surface every 2 s until there was no further change in volume. The loose bulk density (*LBD*) and tapped bulk density (*TBD*) data were calculated using Equations (2) and (3).
(2)$$LBD=\frac{weight\;of\;the\;powder}{volume\;of\;the\;packing}$$
(3)$$TBD=\frac{weight\;of\;the\;powder}{tapped\;volume\;of\;the\;packing}$$

#### 3.5.3. Compressibility Index

Carr’s Index [23] was used to determine the compressibility index of granules, and was calculated using Equation (4).
(4)$$Carr’s\;Index(\%)=\frac{TBD-LBD}{TBD}\times 100\%$$

### 3.6. Evaluation of Tablets

#### 3.6.1. Weight Variation Test

An electronic balance was used to weigh 20 tablets that were taken randomly from each formulation, and the weight values were determined in milligrams (mg) (Mettler Toledo, Oakland, CA, USA).

#### 3.6.2. Hardness Test

Six tablets were taken randomly from each formulation and determined using a hardness tester, with hardness values reported in newtons (N) (Tianda Tianfa Instrument, Tianjin, China).

#### 3.6.3. Friability Test

Approximately 6.5 g of tablets was taken from each formulation and accurately weighed and placed in the friability tester (Xixin Instrument, Tianjin, China), which was set to 100 revolutions for 4 min. After the resulting tablets were dedusted and reweighed, the percentage weight lost was counted as the friability using Equation (5).
(5)$$Friability=\frac{{Weight}_{before\;rotations}-{Weight}_{after\;rotations}}{{Weight}_{before\;rotations}}\times 100\%$$

#### 3.6.4. Content Determination

Twenty tablets were taken at random from each lot and weighed before being placed in a mortar and being porphyrized with a pestle. A quantity equivalent to 5 mg of IMM-H014 (40 mg of powder) was pulled out with 100 mL of hydrochloric acid solution (0.1 N) and supersonically extracted over 30 min. Then, the resulting 10 mL solution was filtered through a polytetrafluoroethylene filter membrane (PTFE, 0.45 μm pore size, Jinteng, Tianjin, China). A C18 column was used in the stationary phase. The collected samples (20 μL) were analyzed using HPLC (Shimazu LC-20AT, Kyoto, Japan) with a UV detector wavelength of 230 nm. IMM–H014 was separated under a mobile phase consisting of ammonium formate PBS (pH 4.0) to acetonitrile at a 60:40 (*v*/*v*) ratio using a C18 column (4.6 mm × 25 cm, 5.0μm). The flow velocity was 0.7 mL/min and the column temperature was 35 °C. The linearity of IMM-H014 ranged from 20.01 μg/mL to 60.02 μg/mL. The validation was conducted following the ICH guidelines.

#### 3.6.5. In Vitro Release Studies

A USP type II dissolution apparatus was used to test the in vitro release of IMM–H014 tablets. The dissolution profiles of five dissolution media were surveyed. The first dissolution medium was 700 mL of 0.01 N HCl (pH 2.0). At 2 h, 200 mL of a 5.25 g/L (*m*/*v*) sodium phosphate solution was added to adjust the pH of the media to 7.5. The other dissolution mediums were a 0.01 N HCl solution, pH 4.5 PBS, water, and pH 6.8 PBS, where the volume of the dissolution medium was 900 mL; the speed was 50 rpm, and and the temperature was maintained at 37 ± 0.5 °C throughout all experiments. The drug release was quantified by ultraviolet–visible spectrophotometry at a wavelength of 230 nm. As a result, different dynamic equations were in accordance with the data of the first media, including zero-order, first-order, Higuchi, and Ritger–Peppas models.

The similarity factor (f_2_) was calculated to compare the release profiles of the IMM–H014 ER tablets during the production and stability study periods, which were defined by Equation (6) [24].
(6)$$f_{2}=50lg{\left\{\left[1+\frac{1}{n}\sum_{t=1}^{n}{\left(R_{t}-T_{t}\right)}^{2}\right]\times 100\right\}}^{-0.5}$$
where *n* is the number of sampling times, and *R_t_* and *T_t_* are the dissolution values of the reference and test samples at each timing, respectively [24].

#### 3.6.6. Stability Studies

The IMM–H014 tablets were packed in aluminum–aluminum blister packaging and placed under the acceleration conditions (40 °C, RH75%) for 6 months. The similarity factor (f_2_) of the dissolution curve, appearance, and content were used as the index of inspection.

### 3.7. Drug Release Mechanism

The following mathematical models with different equations were used to analyze the description of in vitro dissolution [25].
Zero order model: M_t_/M_∞_ = k_0_ t
First order model: ln (1 − M_t_/M_∞_) = −k_1_t
Higuchi model: M_t_/M_∞_ = k_h_ t^1/2^
Ritger–Peppas model: M_t_/M_∞_ = k_k_ t^n^
where the drug release amount at time t is M_t_, and the final drug release amount is M_∞_. The zero-order release rate constant is k_0_; the first-order release rate constant is k_1_; and k_H_ is the zero-order release rate constant [25].

In the Ritger–Peppas model, the diffusion mechanism is expressed by the value of *n*, with *n* ≤ 0.45 corresponding to a Fickian diffusion mechanism, 0.45 < *n* < 0.89 corresponding to a diffusion and erosion skeleton common action mechanism, and *n* ≥ 0.89 corresponding to an erosion skeleton mechanism [26]. For every model considered, the best fit is indicated by the correlation coefficient (r), where an r closer to 1 indicates a better fitting effect.

### 3.8. In Vivo Pharmacokinetics Study in Beagles

#### 3.8.1. Study Design

The study was conducted to compare the pharmacokinetics of the IMM-H014 ER tablets to those of the IMM-H014 IR tablets, following the administration of single doses equivalent to 75 mg (three tablets per dose) in a two-treatment, two-period (time interval was 7 days) crossover design. Ten healthy beagle dogs of either sex were selected for the experiment and randomly divided into two groups (five dogs in each). Before the study, the dogs were fasted for approximately 12 h, with water provided freely. The sampling time points were different due to the different release rates of the ER and IR preparations. Venous blood samples (1–2 mL) were withdrawn into heparinized tubes 0, 0.08, 0.17, 0.25, 0.33, 0.5, 0.75, 1, 2, 3, 5, 8, 12, 24, 36, and 48 h after administration of the IR preparation, while the sampling times for the ER tablets were 0, 0.25, 0.5, 1, 2, 3, 5, 8, 12, 24, 36, and 48 h after administration. The blood samples were promptly centrifuged at 4 °C and 4000 rpm for 10 min to isolate the plasma, which was stored at −20 °C until analysis. A validated HPLC–MS analytical technique was developed to estimate the drug concentration in plasma samples, and the pharmacokinetic parameters (C_max_, T_max_, MRT, and AUC) were estimated.

#### 3.8.2. Statistical Analysis

The plasma concentration of IMM–H014 was plotted versus time to exhibit the pharmacokinetic profiles. Statistical analysis was executed using DAS2.0 software (version 2.0, Mathematical Pharmacology Professional Committee, Shanghai, China). The key parameters of pharmacokinetics, such as C_max_, T_max_, MRT, and AUC, were analyzed. Statistical significance was defined as *p* < 0.05. The relative bioavailability (F) was calculated using the AUC_0–t_ of IR and ER tablets [27].

### 3.9. IVIVC

The IVIVC was developed to relate the percentage of in vitro drug dissolution to the percentage of in vivo drug absorption and was used for drug development. Based on a good correlation, the in vivo pharmacokinetic profile can be determined using the in vitro dissolution rate alone.

The fraction of the drug absorbed (*Fa*) was calculated by the Wagner–Nelson equation [28], as shown in Equation (7).
(7)$$Fa=\frac{C_{t}+k\times {AUC}_{0-t}}{k\times {AUC}_{0-\infty }}\times 100\%,$$
where *Fa* is the fraction of the drug absorbed, C*_t_* is the concentration of the drug in the plasma at time point *t*, k is the elimination rate constant, AUC_0–*t*_ is the calculated area under the plasma concentration curve from zero to time t, and AUC_0–∞_ is the calculated area under the plasma concentration curve from time zero to infinity [28]. The percentage of drug absorption (*Fa*) at the specified timing was drawn against the percentage of drug dissolved in vitro at the same timepoint. The pertinence between the in vitro release and in vivo absorption was assessed by the linear regression coefficient (R).

## 4. Conclusions

In this study, IMM-H014 was developed as an ER preparation to solve the problems of a short oral administration half-life, short administration interval, and high-frequency administration of IMM-H014. IMM-H014 ER tablets composed of hydrophilic polymers were produced via a direct powder pressing method. The optimal formulation of IMM-H014 ER tablets was determined to accelerate stability and remained stable over a 6-month period. For the in vitro dissolution studies, the optimal formulation showed an obvious ER compared to the IR preparation. The results from the in vivo pharmacokinetics study in beagle dogs also clearly indicate that the AUC of the IMM-H014 ER tablets was equivalent to that of the IR preparation, the relative bioavailability was 97.9%, the C_max_ demonstrated a decrease of around 1/3, and the T_max_ and MRT were meaningfully lengthened, with a visible ER impression. Moreover, the IVIVC correlation coefficient (R2) was 0.9509, suggesting that the prepared tablets had a good correlation between the in vitro release and pharmacokinetic effect. Therefore, absorption in vivo can be predicted by the release in vitro.

## Figures and Tables

**Figure 1 ijms-24-12328-f001:**
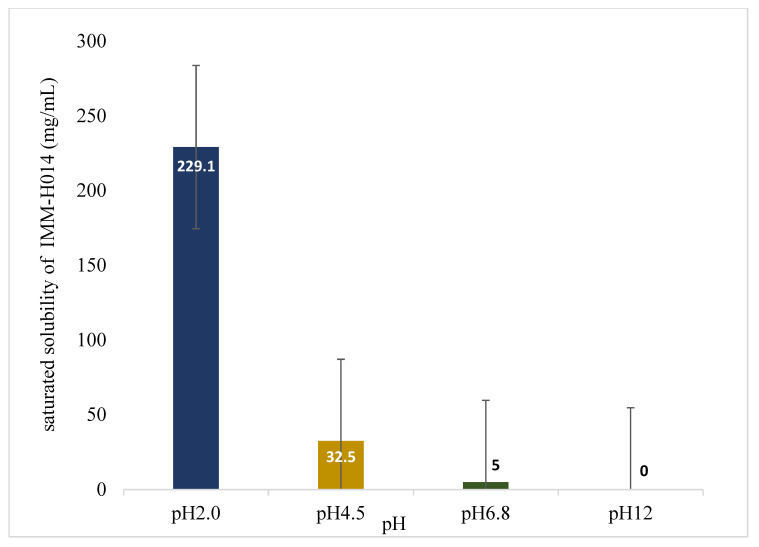
Saturated solubility of IMM-H014 in different media (*n* = 3).

**Figure 2 ijms-24-12328-f002:**
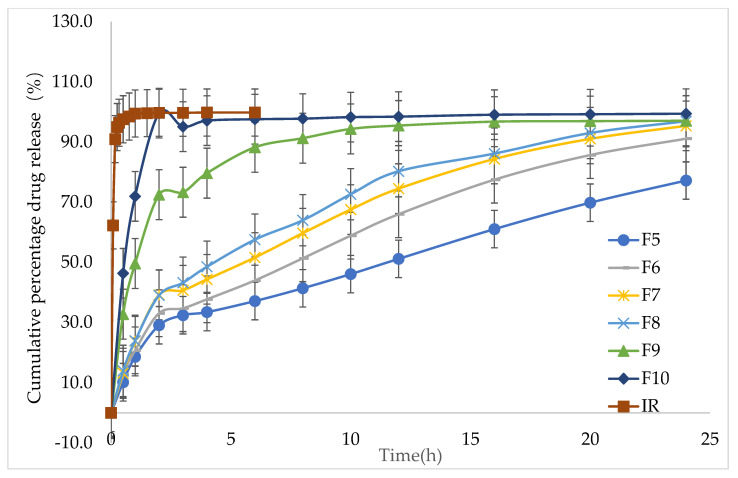
In vitro release profiles of tablets with different HPMC concentrations (immediate release, F5 to F10) in 0.01 N HCl →pH 7.5 PBS medium (*n* = 6).

**Figure 3 ijms-24-12328-f003:**
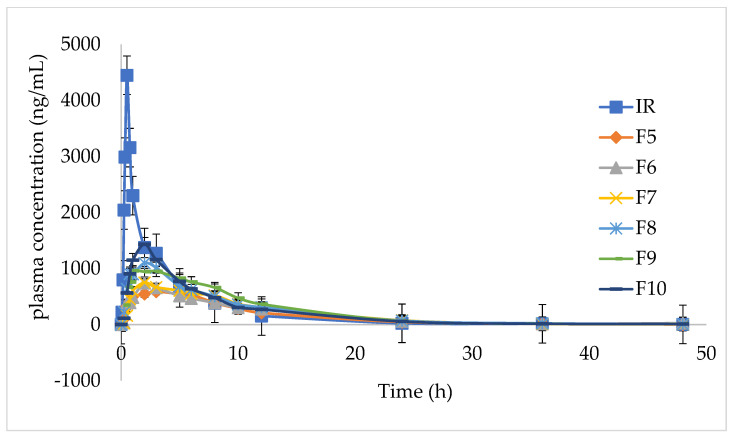
Mean plasma concentrations (ng/mL) of IMM−H014 obtained from administration of ER with different HPMC concentrations and IR to beagles (*n* = 6).

**Figure 4 ijms-24-12328-f004:**
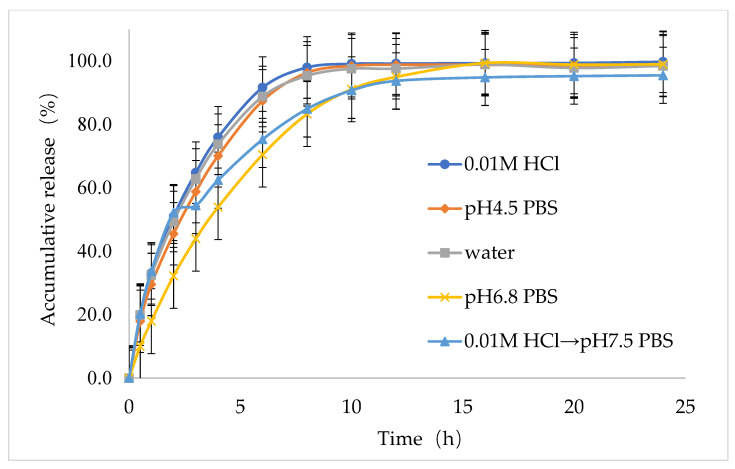
Dissolution profiles of IMM-H014 from extended-release tablets in pH 2.0 HCl, pH 4.5 acetate buffer extraction procedure, pH 6.8 phosphate-buffered saline, water, and pH 2.0 → pH 7.5 phosphate-buffered saline (*n* = 12).

**Figure 5 ijms-24-12328-f005:**
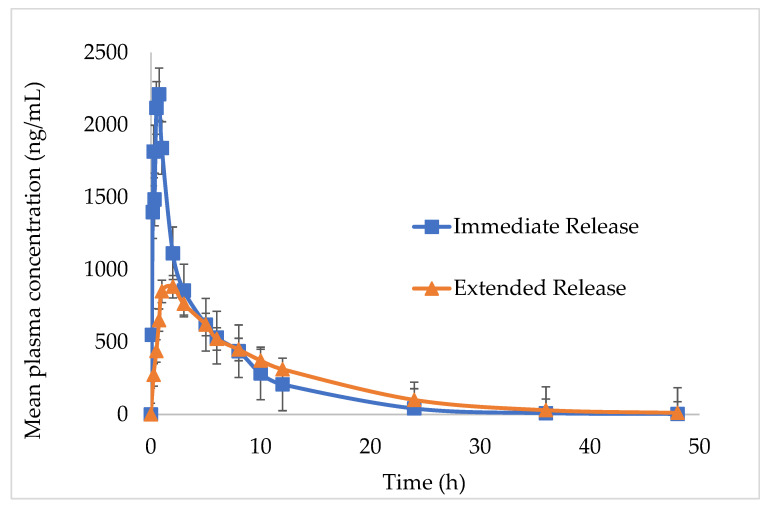
Plasma concentration profiles of the IMM–H014 ER tablet and IR preparation after oral administration to six healthy beagles under fasted conditions (mean ± SD) (*n* = 6).

**Figure 6 ijms-24-12328-f006:**
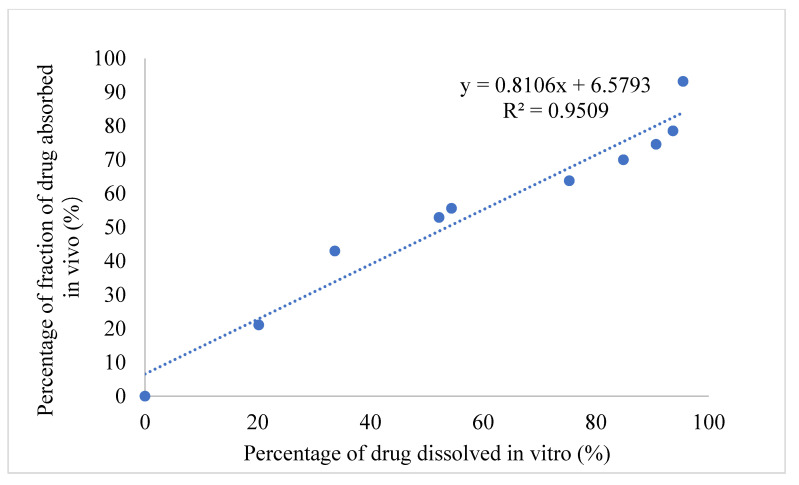
In vivo–in vitro correlation of IMM-H014 ER tablet.

**Table 1 ijms-24-12328-t001:** Composition (in mg) of IMM–H014 tablets.

Code	IR	F1	F2	F3	F4	F5	F6	F7	F8	F9	F10
IMM-H014	25	25	25	25	25	25	25	25	25	25	25
HPMC (90SH–4000SR)	–	60	–	–	–	90	65	50	40	30	20
HPMC (90SH–10,000SR)	–	–	60	–	–	–	–	–	–	–	–
HPC (M–FP)	–	–	–	60	–	–	–	–	–	–	–
HPC (H–FP)	–	–	–	–	60	–	–	–	–	–	–
Lactose	–	111	111	111	111	81	106	121	131	141	151
Silicon dioxide	–	1	1	1	1	1	1	1	1	1	1
Magnesium stearate	–	3	3	3	3	3	3	3	3	3	3
Total weight	25	200	200	200	200	200	200	200	200	200	200

**Table 2 ijms-24-12328-t002:** Physical properties of granules and tablets with different matrix materials.

Code	Angle of Repose (°) of Granules	CI (%) of Granules	Hardness (N) of Tablets	Friability (%) of Tablets	Weight Variation (%) in Tablets
F1	26	13.4	100–130	0.14	−2.57–3.02
F2	27	12.7	100–130	0.15	−3.10–2.89
F3	35	17.5	100–130	0.17	−6.03–5.72
F4	37	18.4	100–130	0.16	−5.82–5.93
F5	30	13.3	100–130	0.18	−2.74–3.59
F6	29	13.0	100–130	0.19	−3.17–3.19
F7	29	12.9	100–130	0.20	−2.98–3.61
F8	27	12.9	100–130	0.17	−3.55–2.14
F9	28	12.7	100–130	0.21	−2.02–4.01
F10	29	12.5	100–130	0.19	−2.19–3.42

**Table 3 ijms-24-12328-t003:** Model fitting result of the in vitro drug release profile of the optimized formulation.

Drug Release Model	Fitted Equation	R^2^
Zero-order	Q_t_ = 5.92t + 33.73	0.87718
First-order	Ln (100 − Q_t_) = −0.36t	0.96333
Higuchi	Q_t_ = 26.42t^1/2^ + 9.26	0.96458
Ritger–Peppas	Qt = 32.87t^0.45^	0.97521

**Table 4 ijms-24-12328-t004:** In vitro release results: IMM-H014 in five types of dissolution media; three batches of IMM-H014 ER tablets (*n* = 12).

Medium	Batch Nos.	f_2_
0.01 N HCl → pH7.5 PBS	1	/
	2	97
	3	74
0.01 N HCl	1	/
	2	86
	3	75
pH 4.5 ABEP	1	/
	2	98
	3	98
water	1	/
	2	81
	3	64
pH 6.8 PBS	1	/
	2	91
	3	61

f_2_ is the similarity factor of the in vitro release profiles of IMM-H014 ER tablets in different media, while batch 1 served as reference.

**Table 5 ijms-24-12328-t005:** Results of accelerated condition stability (40 °C, RH75 ± 5%) test (*n* = 12).

Batch Nos.	Time (Month)	Content (%)	f_2_				
			0.01 M HCl → pH7.5	0.01 M HCl	pH 4.5	Water	pH 6.8
1	0	99.1	/	/	/	/	/
	6	99.2	81	97	96	90	81
2	0	100.2	/	/	/	/	/
	6	99.4	84	74	93	89	79
3	0	100.1	/	/	/	/	/
	6	99.5	94	83	72	80	75

f_2_ is the similarity factor of the in vitro release profiles of IMM–H014 ER tablets at 6 months compared to 0 months.

**Table 6 ijms-24-12328-t006:** Pharmacokinetic parameters of IMM–H014 ER tablets and IR preparation (*n* = 6).

Pharmacokinetic Parameters	Immediate-Release Preparation	Extended-Release Tablet	*p*-Value
AUC(0–t) (ng·h/m)	10,277.3 ± 2212.5	10,062.6 ± 2272.4	0.87
AUC(0–∞) (ng·h/m)	10,291.2 ± 2215.7	10,203.3 ± 2381.6	0.95
MRT(0–t) (h)	6.2 ± 0.8	9.8 ± 1.7	0.001
MRT(0–∞) (h)	6.3 ± 0.9	10.4 ± 2.1	0.001
Tmax (h)	0.7 ± 0.3	2.5 ± 1.4	0.001
Cmax (ng/mL)	3106.9 ± 953.8	1068.0 ± 3.3.4	0.001
F (%)	/	97.9	

Statistically significant difference (*p* < 0.05).

## Data Availability

The data presented in this study are available upon request from the corresponding author.

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
