# Peer review of "Development and In Vitro–In Vivo Correlation Evaluation of IMM-H014 Extended-Release Tablets for the Treatment of Fatty Liver Disease"

_ijms, 2023, doi:10.3390/ijms241512328_

Round 1

Reviewer 1 Report

General observation

It is well known in pharmacology that the intestinal bioavailability (or intestinal passage) of the drug depends on the amount of the active ingredient dissolved at the site of passage and also on the biological membrane. You have confirmed this known principle in your results (relationship between Figure 2 and Figure 3) Figure 6 confirms this old concept. The problem that is often encountered is that the solutions used in vitro in dissolution tests are often not physiological solutions similar to the environment of the intestinal mucosa. Many of the dissolution media used in galenic are solutions containing organic solvents that are prohibited in the development of medicinal products for human and animal use with the exception of ethanol. I am particularly pleased that your study has respected this principle of using physiological solutions compatible with the environment of the intestinal environment. This reinforces the credibility of your study and takes us away from the problems of crystalline polymorphism.

According to the authors, the bioavailability of IMM-H014 is very good in rats (96.3%) (7). This kind of drugs does not cause problems for their use. However, drugs with low bioavailability may be more appropriate for these correlation studies since dosage forms are always being sought to improve their bioavailability of the oral form.

The real challenge in galenic today is not the dissolution - bioavailability relationship, but how to improve the transmembrane passage of drugs with low bioavailability. These aspects will have to be discussed in the discussion if necessary.

Author Response

Dear reviewer,

Thanks for your encouragement and suggestion, we hope this paper will be of interest for readers.

Point 1: According to the authors, the bioavailability of IMM-H014 is very good in rats (96.3%) (7). This kind of drugs does not cause problems for their use. However, drugs with low bioavailability may be more appropriate for these correlation studies since dosage forms are always being sought to improve their bioavailability of the oral form.

Response 1: The results of our group’s previous studies showed that IMM-H014 has good bioavailability, but has a short plasma drug concentration peak time (0.5 h) and short plasma elimination half−life (rats 1.8 h, Beagle dog 3.5 h). Therefore, we designed it as an extended-release formulation to avoid frequent dosing and improve patient compliance. As expected, the IMM–H014 extended-release tablets showed an obvious extended-release effect compared to its immediate-release tablets (see Discussion section 2.7 for details).

Point 2: The real challenge in galenic today is not the dissolution - bioavailability relationship, but how to improve the transmembrane passage of drugs with low bioavailability. These aspects will have to be discussed in the discussion if necessary.

Response 2: Many thanks for your valuable suggestions. Since our research was carried out on IMM-H014, which does not have low bioavailability, transmembrane passages were not discussed. In general, extended-release formulations generally do not improve the bioavailability of a drug, but can be used in conjunction with other techniques (e.g., reduced API particle size, solid dispersions, etc.) to improve bioavailability while maintaining a smooth drug release.

Additional revisions:

  • The source of the three batches of IMM-H014 dispersible tablets have been added in section 3.1 "Materials", as detailed in lines 245-246 of the revised manuscript.
  • According to the requirements of the Preprints.org., the animal ethical information has been added in section 3.1 “Materials”, as detailed in lines 254-258 of the revised manuscript.

If any questions were required, please email to:  yqy@imm.ac.cn

Yours sincerely,

Qingyun Yang  

July 26, 2023

Reviewer 2 Report

The manuscript "Development and in vitro−in vivo correlation evaluation of IMM-H014 extended–release tablets for the treatment of fatty liver disease" is concerning a development of extended–release tablets. This manuscript is well written, although there are several questions/comments that I would like to address to the authors:

1. What is the difference between Figure 1 and Table S1? Why did you implement two approaches to visualize the same data?

2. What different media were used in the solubility experiment? I found only 0.1 N HCl, mentioned in section 3.6.4, although there were media up to pH 12 in Figure 1.

3. Please change the order of the tables. Table 6 should not be the first table mentioned in the manuscript (p. 3, line 96).

4. Change the legend location in Figure S1 so that it does not overlap with one of the release profiles.

5. Please provide standard deviations in your Figures (1-4,6,S1,S2)/Tables (5,S1-S4).

6. What are batches TB2-1, TB2-2 and TA2-1? Does it make sense for potential readers to know exactly how you name the batch? Could it be just 1, 2, 3? Presumably, you do not need to specify the batch at all. You just need to show the mean + - standard deviation and mention that three batches were tested.

7. Can you provide p-values for data in Figure 5? Presumably in the form of Table in SI file.

8. Please improve the figures as the axis labels on all of them are superimposed on the axis divisions, making it difficult for potential readers.

9. Check that all abbreviations you use are spelled out  upon first mention in the manuscript.

Author Response

Dear reviewer,

Thanks for your encouragement and suggestion, we hope this paper will be of interest for readers.

Our article “Development and in vitro−in vivo correlation evaluation of IMM-H014 extended–release tablets for the treatment of fatty liver disease” (ijms-2516882) has been revised in accordance with your suggestion, and we also read it carefully to correct the mentioned errors. The contents of the revision and the response to the reviewer’s comments are described in detail in the revised manuscript.

Point 1:What is the difference between Figure 1 and Table S1? Why did you implement two approaches to visualize the same data?

Response 1: Both Figure 1 and Table S1 show the solubility of IMM-H014 in different media, where Figure 1 in the manuscript uses a bar chart to visualize the difference more clearly, while the specific solubility data are displayed in Table S1 in the Supplementary file.

Point 2: What different media were used in the solubility experiment? I found only 0.1 N HCl, mentioned in section 3.6.4, although there were media up to pH 12 in Figure 1.

Response 2:. Four media, pH 2.0, pH 4.5, pH 6.8 and pH 12, were used for the solubility experiment. The results are shown in Figure 1 and section 2.1, and the detailed test method was described in section 3.2 “Solubility test”. In section 3.6.4 “Content determination”, the HPLC method was employed for the content determination, where 0.1 N HCl solution was used as the solvent.

Point 3: Please change the order of the tables. Table 6 should not be the first table mentioned in the manuscript (p. 3, line 96).

Response 3: Following your useful suggestions, we have updated the numbering of all tables in the order in which they appear in the manuscript (marked in blue font in the revised manuscript).

 Point 4: Change the legend location in Figure S1 so that it does not overlap with one of the release profiles.

Response 4: The legend location in Figure S1 has been moderately adjusted in accordance with the recommendations.

 Point 5: Please provide standard deviations in your Figures (1-4,6,S1,S2)/Tables (5,S1-S4).

Response 5: In accordance with your comments, the standard deviations have been added in Figures 1~4 and Figure S1~S2, sa well as Table 6 (original Table 5) and Table S1~S4, and the details are presented in the corresponding parts of the revised manuscript and Supplementary materials, and the standard deviations in the tables are remarked in blue. In addition, the data in Figure 6 do not involve standard deviations and have not been supplemented.

Point 6: What are batches TB2-1, TB2-2 and TA2-1? Does it make sense for potential readers to know exactly how you name the batch? Could it be just 1, 2, 3? Presumably, you do not need to specify the batch at all. You just need to show the mean + - standard deviation and mention that three batches were tested.

Response 6: TB2-1, TB2-2 and TA2-1 are manufacturer's naming conventions and have no potential meaning. Based on the reviewer’s suggestions, the batch numbers have been simplified to No. 1, 2 and 3 and have been revised accordingly in the revised manuscript.

Point 7: Can you provide p-values for data in Figure 5? Presumably in the form of Table in SI file.

Response 7: In view of your suggestion, the p-values for data in Figure 5 have been supplemented as detailed in Table 6.

Point 8: Please improve the figures as the axis labels on all of them are superimposed on the axis divisions, making it difficult for potential readers.

Response 8: Following your suggestions, we have optimized all the figures in the revised manuscript and supplementary materials. If further optimization is needed, please let us know and we will continue to make improvements.

Point 9: Check that all abbreviations you use are spelled out upon first mention in the manuscript.

Response 9: All abbreviations in the manuscript have been carefully checked and revised, for detailed, see blue font in lines 48-49.

Additional revisions:

  • The source of the three batches of IMM-H014 dispersible tablets have been added in section 3.1 "Materials", as detailed in lines 245-246 of the revised manuscript.
  • According to the requirements of the Preprints.org., the animal ethical information has been added in section 3.1 “Materials”, as detailed in lines 254-258 of the revised manuscript.

If any questions were required, please email to:  yqy@imm.ac.cn

Yours sincerely,

Qingyun Yang

July 26, 2023

Round 2

Reviewer 2 Report

During the review process, I discovered that not all of the previous comments had been answered. Thus, there are a few comments that I would like answers to:

1. What different media were used in the solubility experiment? You have not yet indicated the chemical composition of these media.

2. How can saturation solubility take negative values (Figure 1)? Please, change y-axis.

3. In section 3.2 you have stated that "The amount of the dissolved drug was quantified by HPLC as described in Section 3.6.4". Then, in section 3.6.4. "The linearity for IMM–H014 ranged from 20.01 μg/mL to 60.02 μg/mL." However, the saturation solubilities of the drug in Table S1 were 229.1±8.7, 32.5±1.1, 5.0±0.2 mg|mL. How do these statements correlate with each other? 

 4. Please, indicate how many measurements were taken to calculate the standard deviations in the figures and tables?

Author Response

Dear reviewer,

Thanks again for your suggestion. Our article (ijms-2516882) has been revised based on your suggestion, and we have read it carefully to correct the mentioned errors. Following is a list of revised contents and the response to your comments (blue font in the revised manuscript).

Point 1:What different media were used in the solubility experiment? You have not yet indicated the chemical composition of these media.

Response 1: Four different media were used for the solubility experiments, namely, pH 2.0 hydrochloric acid solution, pH 4.5 acetate-buffered solution, pH 6.8 phosphate-buffered solution, and pH 12 sodium hydroxide solution. In accordance with your suggestion, the preparation methods for various media have been added in section 3.2.1 "Solubility test". See lines 261-272 of the revised manuscript for details.

Point 2: How can saturation solubility take negative values (Figure 1)? Please, change y-axis.

Response 2: Thanks for your criticisms and corrections. The y-axis in Figure 1 has been modified.

Point 3: In section 3.2 you have stated that "The amount of the dissolved drug was quantified by HPLC as described in Section 3.6.4". Then, in section 3.6.4. "The linearity for IMM–H014 ranged from 20.01 μg/mL to 60.02 μg/mL." However, the saturation solubilities of the drug in Table S1 were 229.1±8.7, 32.5±1.1, 5.0±0.2 mg|mL. How do these statements correlate with each other?

Response 3: The test solutions for the solubility experiments were diluted so that the concentration was in the range of 20 μg/mL to 60 μg/mL, which was then determined by the HPLC method under section 3.6.4. In the light of your suggestion, the corresponding description has been added in section 3.2.2, as detailed in lines 276-278 of the revised manuscript.

 Point 4: Please, indicate how many measurements were taken to calculate the standard deviations in the figures and tables?

Response 4: According to your comments, the number of measurements used to calculate the standard deviation have been supplemented with a labeled figure and table in the form of "n=6". For details, see the blue font in the revised version

If any questions were required, please email to:  yqy@imm.ac.cn

Yours sincerely,

Qingyun Yang

July 27, 2023
